# A New Efficient Multi-Object Detection and Size Calculation for Blended Tobacco Shreds Using an Improved YOLOv7 Network and LWC Algorithm

**DOI:** 10.3390/s23208380

**Published:** 2023-10-11

**Authors:** Kunming Jia, Qunfeng Niu, Li Wang, Yang Niu, Wentao Ma

**Affiliations:** College of Electrical Engineering, Henan University of Technology, Zhengzhou 450000, China; kunming_jia@163.com (K.J.); niuqunfeng@haut.edu.cn (Q.N.); yang_niu2023@163.com (Y.N.); wentaoma_2023@163.com (W.M.)

**Keywords:** blended tobacco shred, multi-object detection, size calculation, YOLOv7, LWC

## Abstract

Detection of the four tobacco shred varieties and the subsequent unbroken tobacco shred rate are the primary tasks in cigarette inspection lines. It is especially critical to identify both single and overlapped tobacco shreds at one time, that is, fast blended tobacco shred detection based on multiple targets. However, it is difficult to classify tiny single tobacco shreds with complex morphological characteristics, not to mention classifying tobacco shreds with 24 types of overlap, posing significant difficulties for machine vision-based blended tobacco shred multi-object detection and unbroken tobacco shred rate calculation tasks. This study focuses on the two challenges of identifying blended tobacco shreds and calculating the unbroken tobacco shred rate. In this paper, a new multi-object detection model is developed for blended tobacco shred images based on an improved YOLOv7-tiny model. YOLOv7-tiny is used as the multi-object detection network’s mainframe. A lightweight Resnet19 is used as the model backbone. The original SPPCSPC and coupled detection head are replaced with a new spatial pyramid SPPFCSPC and a decoupled joint detection head, respectively. An algorithm for two-dimensional size calculation of blended tobacco shreds (LWC) is also proposed, which is applied to blended tobacco shred object detection images to obtain independent tobacco shred objects and calculate the unbroken tobacco shred rate. The experimental results showed that the final detection precision, mAP@.5, mAP@.5:.95, and testing time were 0.883, 0.932, 0.795, and 4.12 ms, respectively. The average length and width detection accuracy of the blended tobacco shred samples were −1.7% and 13.2%, respectively. The model achieved high multi-object detection accuracy and 2D size calculation accuracy, which also conformed to the manual inspection process in the field. This study provides a new efficient implementation method for multi-object detection and size calculation of blended tobacco shreds in cigarette quality inspection lines and a new approach for other similar blended image multi-object detection tasks.

## 1. Introduction

The implementation guidelines set out in Articles 9 and 10 of the WHO Framework Convention on Tobacco Control (FCTC) require manufacturers and importers of tobacco products to disclose the contents of tobacco products to government authorities, including the type of tobacco shred and blending ratio of each type of tobacco shred. Tobacco manufacturers must also have equipment and methods for detecting and measuring tobacco shred components [1,2]. According to the National Bureau of Statistics, China’s cumulative cigarette production for the year 2021 reached 2418.24 billion, a cumulative increase of 1.3%. By May 2022, China’s cigarette production stood at 188.25 billion, up 10.4% year-on-year. At present, the construction of tobacco information technology is an important path to achieving industry transformation and industrial upgrading [3]. The relative proportions of each tobacco shred type (tobacco silk, cut stem, expanded tobacco silk, and reconstituted tobacco shred) impact the smoke characteristics, physical indicators, and sensory quality of cigarettes [4,5]. Therefore, efficient and accurate determination of tobacco shred type and the unbroken tobacco shred rate is essential for intelligent information reform, process quality assurance, and production consistency in the tobacco industry.

Machine vision and deep learning methods have been studied in depth in the field of tobacco component detection. Niu et al. [2] constructed a single tobacco shred dataset with four types by shooting single tobacco shreds and adopting a threshold preprocessing algorithm. The Resnet model was improved by changing the block parameters, multi-scale fusion, and focal loss function. The classification accuracy reached 96.56%. Zhong et al. [6] shot single tobacco shreds from four types of tobacco shreds (tobacco silk, cut stem, expended tobacco silk, and reconstituted tobacco shred) for dataset construction. The loading weight was used to improve the Resnet model for classification of the four types of tobacco shreds. The accuracy and recall rate of the model were both higher than 96%. Niu et al. [7] constructed a single tobacco shred dataset by shooting single tobacco shreds from each of the four types and adopting an image preprocessing algorithm. The VGG model was improved by adding a residual module and global pooling layer instead of a fully connected layer for tobacco shred classification. The improved model reduced the amount of parameters by 96.5% and achieved an accuracy of 95.5%. Liu et al. [8] also constructed a single tobacco shred dataset by shooting four types of tobacco shreds. An improved Resnet model was proposed with an efficient channel attention mechanism, multi-scale fusion, and activation function optimization. The final classification accuracy was 97.23%. Wang et al. [9] constructed a single-target 24-type overlapped tobacco shred dataset by shooting random overlapped tobacco shreds of two types. The Mask-RCNN network was improved with densenet121, U-FPN, and optimized anchor parameters. The COT algorithm was proposed to solve the overlapped area calculation of tobacco shreds. The final segmentation accuracy and recall rate were 89.1% and 73.2%, respectively, and the average area detection rate was 90%.

The above tobacco shred research work either studied only the identification of non-overlapped objects of different types of tobacco shreds with a single target in a single tobacco shred picture or the identification of overlapped objects of different types of tobacco shreds with only a single target in a single tobacco shred picture. Thus, it is difficult to apply the above research work and conclusions in real cigarette quality inspections in the field. In the actual cigarette quality inspection of a tobacco factory, tobacco shred pictures taken online contain complex tobacco shred forms within a single picture, containing both non-overlapped and overlapped tobacco shreds. The different types of these blended tobacco shreds need to be identified, but the component ratio of each type of tobacco shred, the unbroken tobacco shred rate, and other index parameters also need to be calculated in real time to judge the quality of cigarettes. On top of that, the time and accuracy requirements are extremely high. It is a huge challenge to complete all of these tasks efficiently and accurately. At present, there are few studies on multi-object rapid detection of blended tobacco shreds for practical application in the field.

With the development of deep learning algorithms, the single-stage YOLO detection algorithm has been widely used in agriculture [10] and other fields [11,12,13,14] to meet the demand for fast detection of multiple targets in a single image. Chen et al. [15] proposed an improved YOLOv7 model for fast citrus detection for unmanned citrus picking. The model achieved multi-scale feature fusion and lightweighting by adding a small object detection layer, GhostConv, and the CBAM attention module. The final average accuracy reached 97.29% with 69.38 ms prediction time. Sun et al. [16] proposed an improved YOLOv5 model for automatic pear picking when fruits are against a disordered background and in the shade of other objects. CBS of the backbone and the fourth stage’s CBS were replaced with the shuffle block and inverted shuffle block, respectively. The final average detection supervision was 97.6% and the model parameters were compressed by 59.4%. He et al. [17] proposed an output estimation method based on the number of soybean pods for multiple pods in a soybean plant. The YOLOv5 model was improved by embedding a CA attention mechanism and modifying the boundary regression loss function. The actual weight of the pods was predicted using a pod weight estimation algorithm. The average accuracy reached 91.7%. Lai et al. [18] proposed an improved YOLOv7 model for the identification of pineapples with different maturity levels in complex field environments. The model was improved by adding SimAM to improve the feature extraction capability, improving MPConv to optimize the feature loss, and adopting soft-NMS to replace NMS. The accuracy and recall rate were 95.82% and 89.83%, respectively. Cai et al. [19] proposed an improved YOLOv7 model for detecting fake banana stems under complex growth conditions. Focal loss function was adopted to solve category imbalance and classification difficulty. Mixup data augmentation was employed to expand the dataset and further improve the accuracy of the model. The accuracy and prediction time were 81.45% and 8 ms, respectively. Based on an improved YOLOv8 model, Li et al. [20] realized efficient multi-target detection under different target size, occlusion, and illumination conditions. The idea of Bi-PAN-FPN was introduced to improve the neck part of YOLOv8-s and enhance the feature fusion of the model. The results showed that the proposed aerial image detection model obtained obvious effects and advantages. Lou et al. [21] proposed an improved YOLOv8 model for small-size target detection in special scenarios. A new downsampling method that could better preserve the context feature information was proposed. Three authoritative public datasets were used in the experiment, all of which improved by more than 0.5%.

For multi-object rapid detection of blended tobacco shreds for practical application in the field, this study proposes an overall solution based on an improved YOLOv7 multi-object detection model and an algorithm for two-dimensional size calculation of blended tobacco shreds to identify blended tobacco shred types and calculate the size of tobacco shreds. The focus is on the identification of blended tobacco shred types, as our research object is determining the best method of identifying tobacco shred components and the unbroken tobacco shred rate for real-world use in quality inspection lines.

For this paper, the main contributions are as follows:(1)Establishing two types of original blended tobacco shred image datasets: 4000 non-overlapped tobacco shreds consisting of images captured from four tobacco shred varieties and 5300 blended tobacco shreds consisting of images captured four non-overlapped tobacco shred varieties and seven types of overlapped tobacco shreds. Dataset 1 was established as the training base model and dataset 2 was used to mimic the actual field and increase the robustness of the model.(2)Developing an accurate YOLOv7-tiny model to achieve multi-object blended tobacco shred detection utilizing digital images. Detection models are developed and compared using Faster RCNN, RetinaNet, and SSD architectures with the chosen datasets. The performances of different tobacco shred detection methods with Resnet50, Light-VGG, MS-Resnet, Ince-Resnet, and Mask RCNN are also compared. The constructed improved YOLOv7 network (ResNetxt19, SPPFCSPC, decoupled head) demonstrated the highest detection accuracy. It provided good detection capability for blended tobacco shred images with different sizes and types, outperforming other similar detection models.(3)Proposing a tobacco shred two-dimensional size calculation algorithm (LWC) to be first applied to not only single tobacco shreds, but also to overlapped tobacco shreds. This algorithm accurately detects and calculates the length and width in images of blended tobacco shreds.(4)Providing a new implementation method for the identification of tobacco shred type and two-dimensional size calculation of blended tobacco shreds and a new approach for other similar muti-object image detection tasks.

## 2. Data Collection

The experimental tobacco shreds were blended tobacco shreds (four varieties of tobacco silk, cut stem, expended tobacco silk, and reconstituted tobacco shred) from one batch of a brand, obtained on the blending line. Prior to obtaining the image datasets, the tobacco shreds were stored in a constant temperature and humidity chamber at 25 °C to reduce the effect of external factors on the image data [9].

In order to achieve rapid and efficient acquisition of blended tobacco shred images, an image acquisition device was designed. It consisted of the following components: industrial camera, surface array light source, vibration platform, closed darkroom, and white balance card. The industrial camera was MV-CH250-90TC-C-NF, which had a resolution of 25 megapixels and captured high-quality tobacco shred images even for of large-area tobacco shred collection. The surface array light source used was MV-LBES-H-250-250-W. Compared with other light sources, it greatly ensured that the brightness of the shooting field of view was uniform, excluding the influence of tobacco shred shadow. A 320 mm × 320 mm laboratory vibration platform was used to avoid the dispersion effect for a single 0.6 g cigarette. A white balance card was used with an 18 degree calibration exposure white board, which met the exposure and color balance needs of the benchmark image. The enclosed darkroom was made of four aluminum plates, which avoided the influence of external light on image acquisition. The overall blended tobacco shred image acquisition device is shown in Figure 1.

Considering the diversity and complexity of dataset construction, 11 types of tobacco shred images were shot, consisting of pure tobacco silk, pure cut stem, pure expended tobacco silk, pure reconstituted tobacco shred, tobacco silk-cut stem, tobacco silk-expended tobacco silk, tobacco silk-reconstituted tobacco shred, cut stem-expended tobacco silk, cut stem-reconstituted tobacco shred, expended tobacco silk-reconstituted tobacco shred, and tobacco silk-cut stem-expended tobacco silk-reconstituted tobacco shred. The four types of pure tobacco shreds were shot to enhance later feature learning. Six types of two blended tobacco shred varieties were shot to enhance the diversity and complexity of the datasets. The final type of blended tobacco shred was shot to simulate tobacco shred distribution in the real field situation. The specific dataset construction sources, quantities, and sample images of 11 types of tobacco shred samples are shown in Table 1 and Figure 2.

Based on the different shot objects, two datasets were constructed in this study. Dataset 1 contained only four types of pure tobacco shred images (Y-pure tobacco silk, G-pure cut stem, P-pure expended tobacco silk, Z-pure reconstituted tobacco shred), with a total of 4000 images for training of the baseline model. Dataset 2 contained all of the samples in dataset 1 and added seven additional types of overlapped tobacco samples (tobacco silk-cut stem, tobacco silk-expended tobacco silk, tobacco silk-reconstituted tobacco shred, cut stem-expended tobacco silk, cut stem-reconstituted tobacco shred, expended tobacco silk-reconstituted tobacco shred, and tobacco silk-cut stem-expended tobacco silk-reconstituted tobacco shred), respectively, for a total of 6800 samples, to enhance the generalization capability of the network model and to cope with the overlapped tobacco shred situation in real applications. Both datasets 1 and 2 were divided into training and testing sets at a ratio of 7:3. A total of 2800 and 1200 samples were used for the training and testing sets from tobacco shred dataset 1, respectively (see Table 2). A total of 4760 and 2040 samples were used for the training and testing sets from tobacco shred dataset 2, respectively (see Table 3). In Table 3, blended Y represents the blended tobacco silk image dataset, blended G represents the blended cut stem image dataset, blended P represents the blended expended tobacco silk image dataset, and blended Z represents the blended reconstituted tobacco shred image dataset.

### 2.1. Data Preprocessing

#### 2.1.1. Background Shadow Elimination Algorithm

Although the image acquisition system ensured efficient and accurate acquisition of blended tobacco shred images to a large extent, the captured tobacco shred images still had problems such as picture background inconsistency and tobacco shred shadows. Therefore, this paper proposes a tobacco shred image background shadow elimination algorithm based on the OpenCV algorithm to optimize image quality, strengthen the foreground features, and keep the image background homogenized. The algorithm flowchart is shown in Figure 3, and the specific steps are:

Step 1—Grayscale processing and expansion treatment: Grayscale the image and perform expansion to expand the foreground information and weaken the background information while preventing foreground information from being lost.

Step 2—Median filtering: The kernel is selected as 3 × 3 to suppress the foreground information and enhance the shadows in the background information.

Step 3—Find the background difference between the original and processed image: Calculate the difference between the original and processed images, with the same bit as black and the tobacco shred as light white. Complete elimination of the shadows in the background with the help of this difference.

Step 4—Background difference normalization treatment: The picture difference value range is normalized to the 0–255 range.

Step 5—Adaptive threshold processing and filtration enhancement: The filter window is adjusted according to the size of the content in the image, and the output image is enhanced.

Figure 4 shows a comparison of the preprocessed results of one blended tobacco shred sample image; Figure 4A is the original unprocessed sample image and Figure 4B is the processed sample image. From Figure 4, it can be seen that the background shadow elimination algorithm proposed in this paper can efficiently remove the shadow interference in the background information and ensure that the texture, color, and morphological information of the tobacco shreds are not distorted.

#### 2.1.2. Data Enhancement

Considering the different morphologies of the four types of tobacco shreds and overlap-type complexity of the blended tobacco shreds, data enhancement was performed in seven ways, including hsv, translate, scale, filplr, mosaic, mixup, and paste_in [22], to extend the dataset sample amount. First, the image was randomly enhanced with hsv in each stage, followed by translation, scaling, and left–right rotation enhancement. Finally, multiple images were blended using mosaic for random stitching and mixup. The specific hyperparameter parameters are shown in Table 4.

#### 2.1.3. Data Annotation

The preprocessed blended tobacco shred images were labeled using labellmg, an image annotation tool. The label types were divided into four types, named Y, G, P, and Z (tobacco silk, cut stem, expanded tobacco silk, and reconstituted tobacco shred, respectively). During the labeling process, non-overlapped tobacco shreds were labeled normally. The labeled images of pure cut stem, as an example, are shown in Figure 5A. For the overlapped tobacco shreds, the process needed to be repeating with multi-labeling, which is shown in Figure 5B, using tobacco silk-cut stem as an example.

## 3. Methods

### 3.1. Overall Framework of the System

The proposed system for multi-object detection of blended tobacco shreds in a large area consists of three main frameworks (image acquisition system, multi-object detection model, and LWC algorithm). First, an image acquisition system is built to perform data acquisition of different types of blended tobacco shreds, and the data are enhanced by homogenization preprocessing and hsv, translate, scale, etc. Second, a new multi-object detection model is developed for blended tobacco shred images based on an improved YOLOv7-tiny model. YOLOv7-tiny is used as the multi-object detection network’s mainframe. The lightweight Resnet19 [23] is used as the model backbone. The original SPPCSPC and coupled detection head are replaced with a new spatial pyramid SPPFCSPC and a decoupled joint detection head [24], respectively. Finally, an algorithm for the two-dimensional size calculation of the blended tobacco shreds (LWC) is also proposed, which is applied to the blended tobacco shred object detection images to obtain independent tobacco shred objects and calculate their lengths and widths for later calculation of the unbroken tobacco shred rate. Figure 6 shows the general framework of multi-object detection of blended tobacco shreds and 2D size calculation of each tobacco shred object.

### 3.2. Proposed Muti-Object Detection Algorithm

In practical applications, the requirement for timeliness is very high when the accuracy rate is certain. Object detection models are mainly divided into single-stage and two-stage detection algorithms. Single-stage detection algorithms are represented by the YOLO and SSD [25] series, while two-stage object detection algorithms mainly include the faster RCNN series. The YOLO series of object detection algorithms are known for their high detection accuracy and fast detection speed and have become the mainstream single-stage object detection algorithms. Therefore, this study is based on the latest YOLOv7 [26] algorithm for multi-object detection of blended tobacco shreds. The YOLOv7 series includes seven different types of models, such as YOLOv7, YOLOv7x, YOLOv7-tiny, etc., to meet the needs of different types and sizes of detection tasks. For a single image containing various complex physical and morphological characteristics and a large number of tiny similar non-overlapped and overlapped tobacco shreds, the multi-object detection task of blended tobacco shreds is challenging, especially the identification of blended tobacco shreds. This paper proposes an improved YOLOv7-tiny tobacco shred multi-object detection algorithm based on the YOLOV7-tiny framework. The following approaches were used to overcome the above challenges:Proposing a new backbone added to YOLOv7-tiny. To enhance the network feature extraction ability of blended tobacco shreds, a lightweight ResNetxt19 is chosen as the backbone network. The novel backbone improvements include the use of multi-scale convolution, cropped and compressed networks, and the connection of different convolutional modules with PANet. Eventually, the network model’s overlapped tobacco shred shallow feature extraction ability is enhanced, and the difficulty of identifying each type of overlapped tobacco shred in blended tobacco shreds is resolved.Replacing the spatial pyramid in the neck structure of YOLOv7-tiny with SPPFCSPC. Optimizing SPPCSPC in YOLOv7 based on the SPPF idea enables faster speedup while keeping the perceptual field unchanged. This solves the problem of model complexity caused by optimizing the backbone network. Lightweighting of the model is further ensured with little impact on the detection accuracy.Replacing the coupled head in YOLOv7-tiny with a decoupled head enables the generation of spatially coarse but more semantically informative feature encoding for the tobacco classification task. Additionally, the prediction of high-resolution feature maps containing more edge information by different convolutional branches for the tobacco localization task greatly optimizes the model performance in detecting blended tobacco shreds and further resolves the non-consistency problem of blended tobacco shred identification and target localization.

The overall framework diagram of the improved YOLOv7-tiny model is shown in Figure 7. Detailed improvements are described in Section 3.2.1, Section 3.2.2 and Section 3.2.3

#### 3.2.1. Improved YOLOv7-Tiny Model—About the Backbone

In YOLOv7-tiny, multi-scale feature extraction of the input image is performed by the convolution and ELAN modules in the backbone. However, the multi-scale ELAN module based on the lightweight design did not effectively perform feature extraction for the complex blended tobacco shreds.

A lightweight Resnet19 is designed as a backbone consisting of four layers to enhance the feature extraction of blended tobacco shreds. The ELAN module has an excellent local expansion structure, i.e., it performs multiple convolution operations on the input image in parallel and stitches all the input results into a very deep feature map. For small-target blended tobacco shreds where shallow information is critical, the shallow information is not effectively exploited by multi-scale feature fusion, although it is possible. The convolutional network with feature reuse structure can effectively extract features for critical shallow information objects. Based on this, the four Re-I-blocks are designed by fusing the multi-scale convolutional ELAN structure and the feature reuse ResNet50 [27], as described in Figure 8. The feature map in the width direction is widened by multi-scale convolution, and the feature map in the depth direction is extended by feature reuse, which finally enhance the feature extraction ability of small-target tobacco shreds. In order to further enhance the network’s ability to extract the shallow information of blended tobacco shreds, the first layer connects to P3 of PANet [28] via CBS and MP, and the second layer connects to P3 of PANet via CBS. The third layer, as P4 of PANet, connects through CBS. However, the multi-scale feature reuse of Resnet50 inevitably increases the network parameters, which is insufficient compared to the original YOLOv7-tiny’s lightweight backbone. Based on this, Resnet50 is lightened by pruning and compressing of network. Subsequent experiments verified that the four modules of Resnet50 were changed from the original 3:4:6:3 to 1:2:2:1 by using the compressed model, and the network feature extraction capability was still maintained with a significant reduction in model parameters. The details are shown in Figure 9.

#### 3.2.2. Improved YOLOv7 Model—About the Neck

Due to the limitation of storage space and power consumption in real devices, lightweighting of network models is a prerequisite for their practical use. SPP [29] is a spatial pyramid pooling structure that can effectively avoid image distortion and repetitive feature extraction in convolutional networks. SPPF is simplified to improve the parallel connection of the Maxpool structure to serial connection, resulting in faster model computation speed and shorter inference time. SPPCSPC in YOLOv7-tiny is developed based on SPP. It not only retains the SPP module to perform multi-scale spatial pyramidal pooling of the input feature maps, thus obtaining multiple feature maps of different scales that can capture target and scene information of different sizes, but also uses the convolution module to perform convolution operations on the output feature maps, thus further improving the feature representation capability. Although SPPCSPC in YOLOv7 can improve multi-object detection accuracy, it has a greater impact on the network inference speed. Considering that Section 3.2.1 deepens the network depth and will further grow the model inference time, the design of SPPCSPC is simplified with the idea of the SPPF module, which greatly improves the inference efficiency and ensures the model is lightweight with little impact on the detection accuracy, as shown in Figure 10.

#### 3.2.3. Improved YOLOv7 Model—About the Head

Although the backbone and FPN of YOLO series have been continuously optimized, their detection heads are still coupled. The coupled detection head is not optimal for detection performance. The fully connected head is more suitable for the classification task (with stronger ability to distinguish between complete and partial targets) and the convolutional head is more suitable for the localization task (with stronger ability to classify targets).

Based on the above improvements to the backbone and neck, the model has richer feature information. In YOLOv7-tiny, multiple tasks are set in the head, including classification, regression, and object detection. Thus, these tasks are not independent. In this paper, the detection head is decoupled, which can optimize the accuracy of the classification task, regression task, and improve object detection. The decoupled head is composed of (1) a 1 × 1 Conv layer to reduce the channel dimension, (2) two parallel branches for decoupling, (3) a convolutional branch for the classification task, and a fully connected branch for the location regression task with an IoU branch added. See Figure 11 for the decoupled head details.

### 3.3. Blended Tobacco Shred Size Calculation Algorithm

Based on the improved YOLOv7-tiny algorithm, we could not only quickly detect multi-object tobacco shreds, but also finish the calibration of their positions. Then, the unbroken tobacco shred rate could be successfully calculated. The proposed blended tobacco shred size calculation algorithm uses the improved YOLOv7-tiny network to calibrate various tobacco shreds, determine their locations, and extract and crop them with their location information. The cropped tobacco images are then removed from the image, and the corresponding lengths and widths are calculated. The algorithm for the size calculation of blended tobacco shreds, named the LWC algorithm, is as follows, and the specific process effect is shown in Figure 12.

Crop the tobacco shred object and eliminate excess tobacco shreds. Firstly, crop the image with the help of the calibration information generated by YOLOv7-tiny. Secondly, grayscale processing, binarization, contour extraction, contour center statistics, and contour filtering are performed for the cropped tobacco shred image, respectively. According to the YOLOv7 algorithm labeling principle, it is known that the cropped image may contain more than one object. The contour center statistics show that the contour corresponding to the contour center point near the position of the center of the marker box is the calibrated tobacco shred object.Pixel conversion. Calculate the standard volume block (actual length of the pixel area is named Length_standard). The calculation formula is as follows:


(1)
$$Pixel\;per\;unit\;length=\frac{Length}{Length\_standard}$$


Length calculation. Introducing the idea of calculus [30] for the bent object, a segmented fit to the center is performed. Starting from the center line of the tobacco shred, the obtained length value is more accurate compared to that obtained by other methods by measuring along the center line of the tobacco shred from beginning to end.Width calculation. By introducing the idea of calculus, the maximum inner joint circle is fitted to the bent object in segments. The width is measured from the beginning to the end of the bend, starting from multiple maximum internal junction circles of the tobacco shred profile.

Specific steps for length calculation:Search for the outline and intercept of the maximum form of the tobacco shred, and then rotate the maximum form so that the rectangle outside the maximum outline of the tobacco is horizontal (the length of the rectangle is x1, the width is y1). The X/Y coordinate system is established with the lower right corner of the rotated rectangle (the longest distance is x1, the highest distance is y1).Using ∆x = x1/n as the distance, multiple segmentation cuts are made to the image of the tobacco shred in the coordinate system.The image of the tobacco shred is cut into m copies (m = x1/∆x). Partial contours are all contained in m images, and the center point of the contour is searched. All of the centroids are found by iterating through m images.Curve fitting is performed for all of the center points, which gives the fitted center curve of the bent tobacco shreds.

Specific steps for width calculation:Search for the outline of the tobacco and intercept of the maximum form of the tobacco shred, and then rotate the maximum form so that the rectangle outside the maximum outline of the tobacco shred is horizontal (the length of the rectangle is x1, the width is y1). The X/Y coordinate system is established with the lower right corner of the rotated rectangle (the longest distance is x1, the highest distance is y1).Using ∆x = x1/n as the distance, multiple segmentation cuts are made to the image of the tobacco shred in the coordinate system.The image of the tobacco shred is cut into m copies (m = x1/∆x). Partial contours are all contained in m images, and the maximum inscribed circle in the contour is searched. The m images are iterated to find all of the maximum inscribed circles.The average width of the bent tobacco shred is calculated by averaging the diameters of all of the maximum inscribed circles.

## 4. Results

### 4.1. Implementation Details

To effectively validate the blended tobacco shred detection performance and timeliness of the network model, precision, recall, mAP@.5, mAP@.5:.95, and parameters in YOLOv7 were used as evaluation indicators for the improved YOLOv7 model and other baseline detection models [31]. Higher values of precision, recall, mAP@.5, and mAP@.5:.95 and a smaller number of parameters are more ideal in the multi-object detection model.

The calculation formula of precision is as follows:(2)$$Precision=\frac{TP}{TP+FP}$$

The calculation formula of recall is as follows:(3)$$Recall=\frac{TP}{TP+FN}$$

The calculation formula of mAP is as follows:(4)$$AP=\int_{0}^{1}p\left(r\right)dr$$
(5)$$mAP=\frac{1}{n}\sum_{i=1}^{n}API$$
where mAP@.5 is the average mAP with a threshold larger than 0.5; mAP@.5:.95 is the average mAP at different IoU thresholds (from 0.5 to 0.95 with steps of 0.05).

Parameters: The number of parameters reflects the size of the resources occupied by the model.

The experimental platform ran on a 64-bit window operating system with an Intel(R) Core (TM) i9-13900KF processor, 64G of RAM, and a NVIDIA GeForce RTX3090Ti graphics card configuration. Model training was performed in the pytorch framework with python programming.

The calculation formula of loss function (CIoU) is as follows:(6)$$IoU=\frac{A\cap B}{A\cup B}$$

A indicates the prediction box and B indicates the real box. The higher the value of IoU, the higher the degree of overlap between boxes A and B, and the more accurate the model prediction.
(7)$$CIoU=1-IoU+\frac{\rho^{2}\left(b,b^{gt}\right)}{c^{2}}+av$$
where *a* is the weight function and *v* measures the consistency of the aspect ratio.
(8)$$a=\frac{v}{\left(1-Iou\right)+v}$$
(9)$$v=\frac{4}{\pi^{2}}{\left(arctan\frac{w_{gt}}{h_{gt}}-arctan(\frac{w}{h})\right)}^{2}$$

### 4.2. Results of the Multi-Object Detection Model

The training and testing were performed for 100 rounds using YOLOv7-tiny and the improved YOLOv7 model with datasets 1 and 2 for tobacco shreds under the configuration of Section 3.1. The experimental results in Table 5 show that (1) the precision and mAP@.5 of YOLOv7-tiny were 0.901 and 0.956, respectively, in dataset 1 with no obvious overlap, indicating that the model could accurately accomplish object detection task. (2) In dataset 2 with blended tobacco shreds, both the precision and mAP@.5 of YOLOv7-tiny showed large decreases, which shows that the model would have more problems in actual use. (3) The precision and mAP@.5 of the improved YOLOv7 model were 0.883 and 0.932, increasing 4.9% and 4.3% compared with YOLOv7-tiny, respectively. This proved the proposed model’s feasibility and it could accurately detect blended tobacco shreds, although the parameters increased by 2.41 times. Based on the Section 4.3.4 experimental data, the number of model parameters was still relatively much smaller compared to other tobacco shred detection algorithms, laying a solid foundation for fast and accurate detection of multi-object blended tobacco shreds in real field applications. The specific identification example diagram is shown in Figure 13.

### 4.3. Ablation Experiment

#### 4.3.1. Performance Improvement Based on Backbone

In this section, the feature extraction ability of the new backbone for blended tobacco shreds is verified. The network performance was compared for training and prediction among 11 models of Regnet, Efficient, MobileNet, Resnet50, and Resnet with different PANet connections, named YOLOv7-tiny-B1-B6, as shown in Figure 14 using dataset 2.

The results of the experiments in Table 6 showed that:(1)Regnet (deep network) [32], Resnet50 (deep network), Efficient (shallow network) [33], and MobileNet (shallow network) [34] were selected for YOLOv7-tiny backbone replacement. The optimal multi-object detection performance was found experimentally with the backbone of Resnet50. The precision and mAP@.5 were 0.864 and 0.913, respectively. They increased by 3% and 2.4% compared with YOLOv7-tiny, proving that the feature extraction ability was enhanced by the feature reuse structure. However, the parameters of the model increased by 366%.(2)Comparing YOLOv7-tiny-Resnet50 with YOLOv7-tiny-B2, it can be seen that both the precision and mAP@.5 of YOLOv7-tiny-B2 (Resnet50) had small increases and there was a small decrease in the number of parameters of the model, demonstrating that the Re-I-block with multi-scale convolution and feature reuse could further enhance the feature extraction ability of blended tobacco shreds.(3)Comparing YOLOv7-tiny-B1, YOLOv7-tiny-B2, and YOLOv7-tiny-B3, we can see that the indexes of YOLOv7-tiny-B2 were the best, proving that the detection results were best with PANet’s P3 and P4 connecting the shallow layer (the second layer of Resnet) and the middle layer information (the third layer of Resnet), respectively.(4)Considering the characteristics of small-target blended tobacco shreds with rich shallow information, based on YOLO-tiny-B2, PANet’s P3 not only connected the second layer of Resnet but also connected the first layer of Resnet by convolution and downsampling processing. The experimental results proved that all of the indexes of YOLOv7-tiny-B4 were further improved.(5)Considering the substantial increase in the number of model parameters for YOLOv7-tiny-B4 compared to YOLOv7-tiny, both pruning and compression were used in order to keep the model lightweight. The pruning approach was that the fourth layer of Resnet was removed to keep the connection of PANet, considering that the first three layers of features of the Resnet model have a large impact on model feature extraction. The compression approach was that the number of blocks in the four layers of Resnet was reduced from 3:6:4:3 to 1:2:2:1.(6)Comparing the results of YOLOv7-tiny-B5 and YOLOv7-tiny-B6 showed that the effect of compressing the model was superior to that of paper-cutting the model in order to make the model lighter. Compressing Resnet50 (3:4:6:3) to Resnet19 (1:2:2:1) enhanced the weight of shallow information (second layer of Resnet) and middle information (third layer of Resnet) and weakened the weight of superficial information (first layer of Resnet) and deep information (fourth layer of Resnet). Compared to YOLOv7-tiny-B4, the amount of parameters of the model was reduced by 49% while the indicators of the model remained unchanged.(7)Finally, the precision, mAP@.5 and parameters of YOLOv7-tiny-B6 compared to YOLOv7-tiny increased by 4.2%, 3.5%, and 120%, respectively.

#### 4.3.2. Performance Improvement Based on Neck

In this section, validation of the model complexity problem posed by SPPFCSPC optimizing the backbone network is carried out based on the improved proof of Section 4.3.1. Dataset 2 was trained and predicted under the above parameter configuration to compare the network performance.

As can be seen in Table 7, the performance of YOLOv7-tiny-B6-SPPFCSPC was basically unchanged compared to YOLOv7-tiny-B6. However, with the same model parameters, the predicted time (ms) of the network model was reduced by 30%, proving the effectiveness of the improvement for real applications and demonstrating that the network performance could be maintained while further optimizing model lightness. YOLOv7-tiny-B6-SPPFCSPC was named YOLOv7-tiny-BS.

#### 4.3.3. Performance Improvement Based on Head

In this section, verification of the effectiveness of the double head with decoupled detection heads for classification and location is performed based on the improved proof of Section 4.3.2. Dataset 2 is trained and predicted under the above parameter configuration to compare the network performance.

As can be seen in Table 8, the performance of YOLOv7-tiny-BS-decoupled-head was optimal in all evaluation metrics, with increases of 0.6%, 0.9%, 2.1%, and 55% in precision, mAP@.5, mAP@.5:.95, and parameters, respectively, proving the improved validity. Although the number of parameters of the model increased, the lightness of the model was ensured by SPPFCSPC processing. The YOLOv7-tiny-BS-decoupled-head model was named Improved YOLOv7-tiny.

#### 4.3.4. Comparison with Other Object Detection Methods

To further demonstrate the superiority of the YOLOv7-tiny and Improved YOLOv7-tiny algorithms for blended tobacco shred detection, Faster RCNN [35], RetinaNet [36], SSD, YOLOv7-tiny, and Improved YOLOv7-tiny were studied for training and predicting using dataset 2 under the above parameter configurations to compare the network performance.

As seen in Table 9, YOLOv7-tiny had the highest precision and mAP@.5 compared to Faster RCNN, RetinaNet, and SSD, which proved that the YOLOv7-tiny network had the best performance for blended tobacco shred multi-object detection. Improved YOLOv7-tiny had the highest precision and mAP@.5 compared to YOLOv7-tiny, although the parameters increased by 241%, which was still less than most mainstream object detection models, but the precision, recall, mAP@.5, and mAP@.5:95 of the network increased by 4.9%, 5.1%, 4.3%, and 5.9%, respectively. From Table 9, it can be concluded that Improved YOLOv7-tiny proposed in this paper could effectively and quickly complete the fast multi-object detection task of blended tobacco shreds.

In order to further validate the practicality and effectiveness of Improved YOLOv7 in practical applications of blended tobacco shred recognition, recognition ideas, objects, and results were compared among previously studied models of Resnet50, Light-VGG, MS-Resnet, Inception-Resnet, Mask RCNN, and Improved YOLOv7. The comparison results of specific recognition objects and effects are shown in Table 10. In Table 10, the “×” sign represents no mention in the literature. The preprocess time represents the time to segment multiple individual objects in one picture. The start-up time represents the time to load the model. The predict time is the model prediction time. The total time is the overall model prediction time, including preprocess time, start-up time, and predict time.

In the above research processes, the main ideas of tobacco shred recognition are divided into: (1) Firstly, a single tobacco shred image containing only multiple single non-overlapped shreds is processed by the segmentation algorithm to establish a single tobacco shred object image. Secondly, the image classification methods are performed on multiple single tobacco shred images using image classification models such as Resnet50, Light-VGG, MS-Resnet, and Inception-Resnet. (2) Firstly, a single tobacco shred image containing multiple overlapped tobacco shreds is processed by a segmentation algorithm to identify overlapped tobacco shreds. Secondly, a segmentation model is used for a single overlapped tobacco shred object to perform image segmentation processing, such as Improved Mask RCNN. (3) In this paper, the idea is to create a blended tobacco shred image that reflects the real field situation as a sample dataset, with one tobacco shred image containing both single and overlapped tobacco shreds, which is directly processed by image tagging without the need to create multiple single tobacco shred images in advance through segmentation algorithms and the corresponding blended tobacco shred detection results are directly output with the multi-object detection model.

As can be seen from Table 10, the results of previous studies have the following three limitations in practical field applications of tobacco shred type recognition: (1) They are unable to achieve multi-object detection in a single tobacco shred image. Tobacco shred classification models such as Resnet50, Light-VGG, MS-Resnet, and Ince-Resnet can realize single object classification of non-overlapped tobacco shreds with single tobacco shred image data; the tobacco shred segmentation model of Mask RCNN achieves single-target segmentation of overlapped tobacco shreds in a single image. (2) It is not possible to simultaneously achieve multi-object detection of both single and overlapped tobacco shreds in a single image. Classification models such as Resnet50, Light-vgg, MS-Resnet, and Ince-Resnet can only achieve non-overlapped tobacco shred classification; the segmentation model of Mask RCNN achieves only overlapped tobacco shred segmentation. (3) Each single tobacco shred object must be created and the prediction time is long. Due to the limitations of the classification and segmentation network, only a single object of a single image can be processed. This results in a cumbersome detection process and long detection times. The minimum difference in detection speed was 9.67 times (between Light-vgg and Improved YOLOv7) and the maximum difference was 21.5 times (between Mask RCNN and Improved YOLOv7).

### 4.4. Evaluation of the Blended Tobacco Shred Size Calculation Algorithm

For evaluation of the blended tobacco shred size calculation algorithm, five samples of each the four varieties of tobacco shreds (tobacco silk, cut stem, expended tobacco silk, and reconstituted tobacco shred) were taken and a total of 20 sets of experiments were carried out to test the algorithm. The experimental tobacco shred sample datasets and size measurement results comparison are shown in Figure 15 and Table 11 and Table 12.

As can be seen from Table 11, Standard represents a standard rectangular block with a length of 3 mm and width of 9 mm. Y1–Y5 represent five samples of tobacco silk, G1–G5 represent five samples of cut stem, P1–P5 represent five samples of expended tobacco silk, and Z1–Z5 represent five samples of reconstituted tobacco shred. The actual lengths and widths of the tobacco shreds were measured manually. Each tobacco shred sample was divided into multiple parts for measurement of its irregularly curved shape. The relative error of the length and width detected by the proposed tobacco shred size calculation algorithm was calculated with total sum of multiple part sizes. It can be seen from the results of the relative error that the G5 sample obtained the best accuracy with −0.3% for length and 3% for width, respectively, while P5 was the worst for length (−14.6%) and G1 was the worst for width (59.9%).

Total-Y, Total-G, Total-P, Total-Z, and Total represent the total samples of tobacco silk, cut stem, expended tobacco silk, reconstituted tobacco shred, and the total 20 samples of tobacco shreds in Table 12, respectively. The relative error for the total samples reached −1.7% for the detected length and 13.2% for the detected width with high accuracy in the tobacco shred size calculation. In the manual measurement method, multi-part sizes need to be measured one by one for the length and width directions of one tobacco shred, which results in measurement difficulties such as the expended tobacco silk shred being fragile due to carbon dioxide processing. The proposed size calculation algorithm showed superiority in accuracy and time for measuring various tiny and complex shapes of tobacco shreds compared with manual measurements adopted in the field. The method meets the demands of practical tobacco shred size measurement.

## 5. Summary

Practical cigarette quality inspection line applications pose significant difficulties for rapid and accurate multi-target detection of blended tobacco shreds and calculating the unbroken tobacco shred rate. This study develops an improved YOLOv7-tiny multi-object detection model with an LWC algorithm to overcome the problems of identifying both tiny single and various overlapped tobacco shreds with complex morphological characteristics at once, that is, fast blended tobacco shred detection based on multiple targets and unbroken tobacco shred rate calculation tasks. This model was successfully applied to the multi-object detection of blended tobacco shreds and unbroken tobacco shred rate calculation to meet practical field needs. Based on the aforementioned statements, the following innovations were achieved:A data collection platform for blended tobacco shreds was built to mimic the actual inspection environment. Datasets based on four conventional, pure, tobacco shreds and blended tobacco shreds were both created and preprocessed for homogenization to solve background inconsistencies, aiming at accurately identifying blended tobacco shreds in the actual production process. A 5300-sample dataset was enhanced with hsv, translate, scale, filplr, mosaic, mixup and paste_in, taking into account different complexity characteristics of blended tobacco shreds, which effectively avoided overfitting and ensured suitability for actual field use.An improved YOLOv7 model is proposed with improvements such as multi-scale convolution, feature reuse, different PANet connections, and compression. Resnet19 as a new backbone of YOLOv7-tiny is constructed to enhance the depth and width of model feature maps without significantly increasing the network parameters. Secondly, SPPCSPC in the neck structure is changed to SPPFCSPC to increase model inference speed without increasing the model parameters. Finally, the head structure is optimized to a decoupled head, which decouples convolution for the detection task to enhance model multi-object detection performance. In the end, comparing different object detection models and different blended tobacco shred detection algorithms, Improved YOLOv7 had the best performance and was able to quickly and accurately complete multi-object detection of blended tobacco shreds in the field.An LWC algorithm is proposed to obtain the unbroken tobacco shred rate of blended tobacco shreds. The LWC algorithm had an accuracy error of −1.7% in the length calculation and 13.1% in the width calculation. The LWC algorithm significantly improves the detection accuracy of the total size of the blended tobacco shreds.

The precision, mAP@.5, parameters, and prediction time of the proposed Improved YOLOv7, which were 0.883, 0.932, 20,572,002, and 4.12 ms, respectively, meet the practical requirements of cigarette inspections in the field in terms of accuracy and timeliness. The proposed system is very promising for actual production processes in terms of multi-object tobacco shred detection and unbroken tobacco shred rate calculation. 

However, this study has several limitations, namely that some of the blended tobacco shreds are missed in the detection process, and the accuracy of tobacco shred detection and width calculation by the LWC algorithm needs to be further improved.

Follow-up work should consider the aspects below:(1)Further optimize the performance of the model according to the missed detection of blended tobacco.(2)Optimize the LWC algorithm to minimize width measurement error of tobacco shreds and overcome severe curvature reflection.(3)The scheme proposed in this paper must be installed and applied in an actual tobacco quality inspection line to further optimize and verify the model and LWC algorithm.

## Figures and Tables

**Figure 1 sensors-23-08380-f001:**
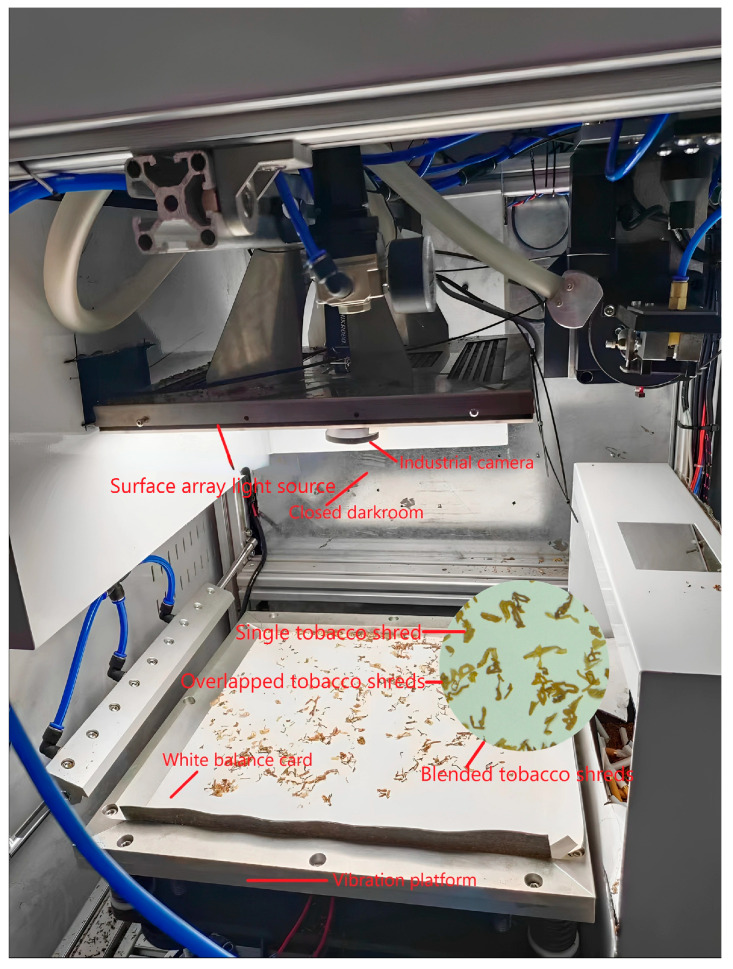
Tobacco shred image acquisition device.

**Figure 2 sensors-23-08380-f002:**
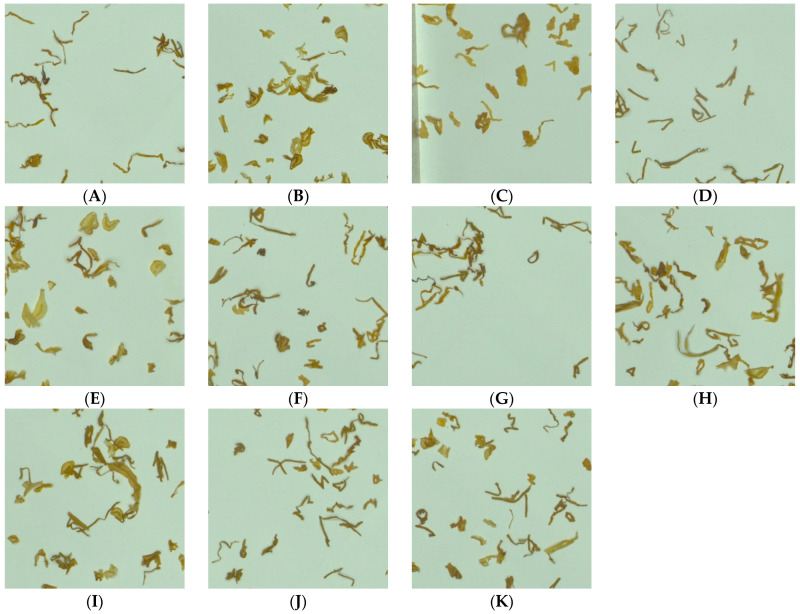
Eleven types of tobacco shreds. (**A**) Pure tobacco silk. (**B**) Pure cut stem. (**C**) Pure expended tobacco shred. (**D**) Pure reconstituted tobacco shred. (**E**) Tobacco silk-Cut stem. (**F**) Tobacco silk-Expended tobacco silk. (**G**) Tobacco silk-Reconstituted tobacco shred. (**H**) Cut stem-Expended tobacco silk. (**I**) Cut stem-Reconstituted tobacco shred. (**J**) Expended tobacco silk-Reconstituted tobacco shred. (**K**) Tobacco silk-Cut stem-Expended tobacco silk-Reconstituted tobacco shred.

**Figure 3 sensors-23-08380-f003:**
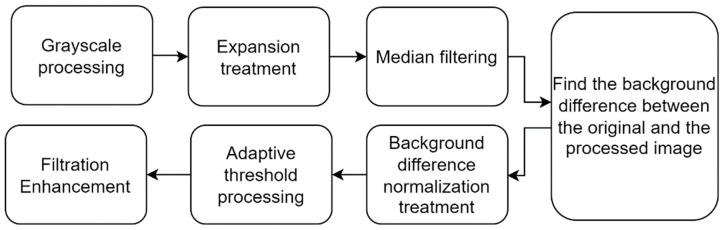
Background homogenization process.

**Figure 4 sensors-23-08380-f004:**
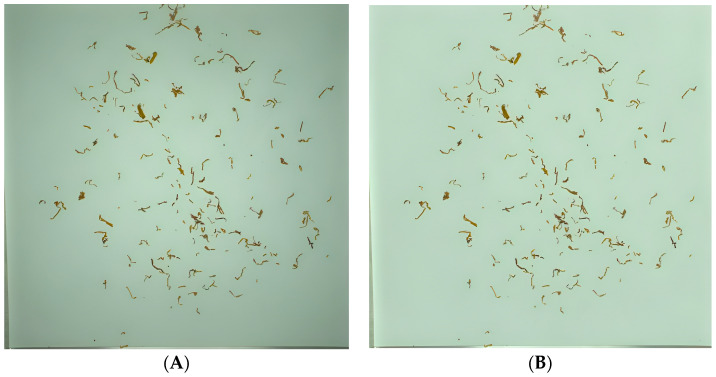
Preprocessing effect comparison. (**A**) Original picture. (**B**) Processed image.

**Figure 5 sensors-23-08380-f005:**
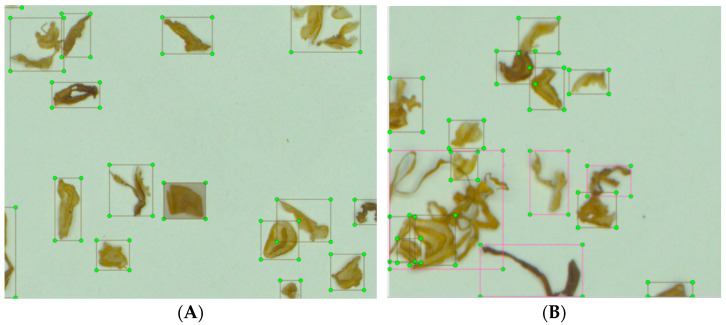
Labeled tobacco shred images. (**A**) Example of pure cut stem. (**B**) Example of tobacco silk-cut stem.

**Figure 6 sensors-23-08380-f006:**
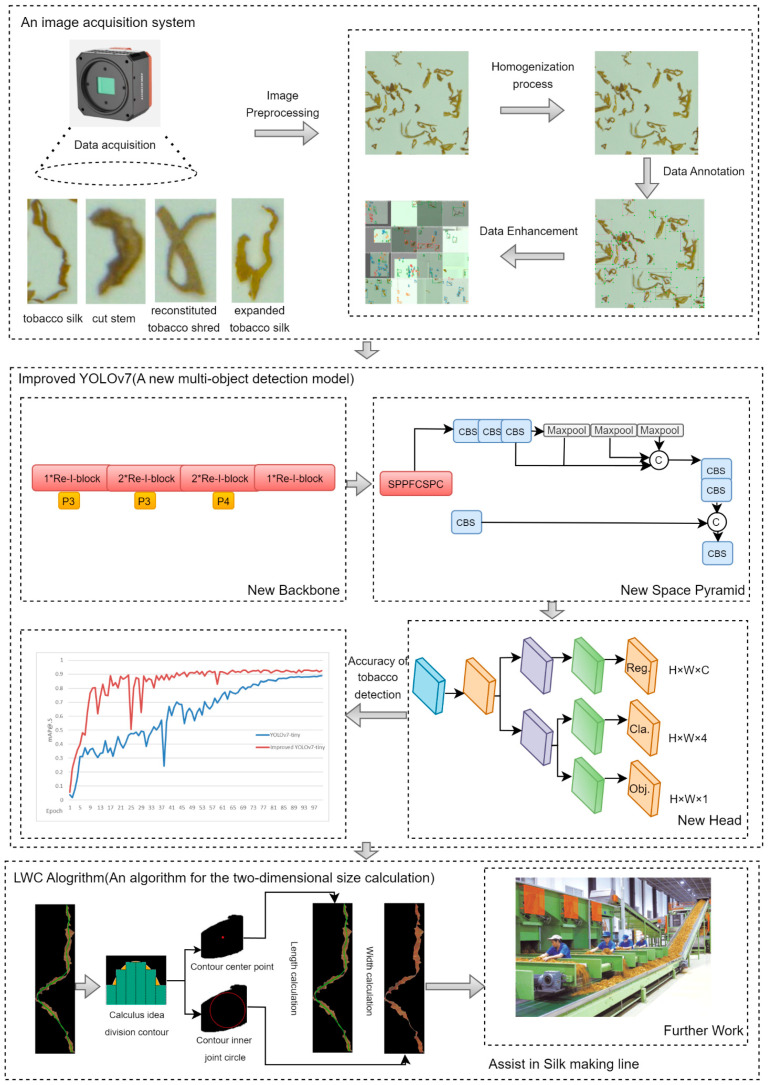
Overall framework of system.

**Figure 7 sensors-23-08380-f007:**
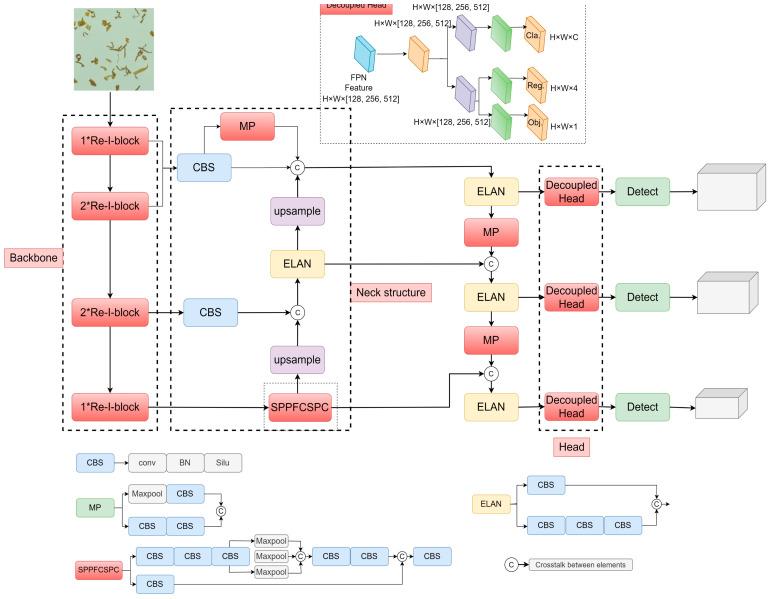
Overall framework of improved YOLOv7.

**Figure 8 sensors-23-08380-f008:**
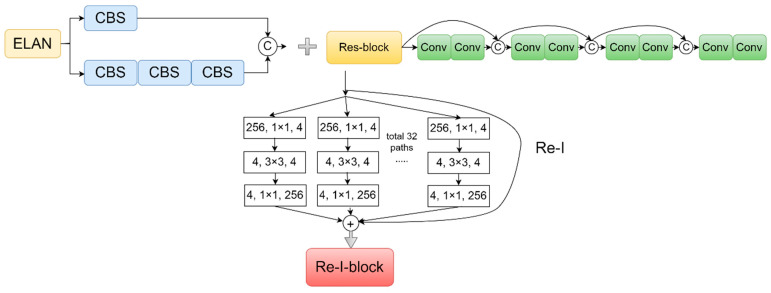
The Re-I-block architecture.

**Figure 9 sensors-23-08380-f009:**
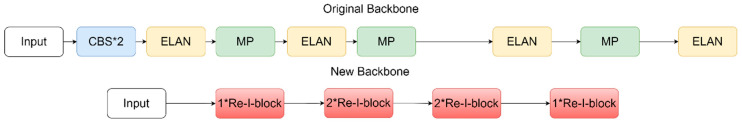
Original and new backbones.

**Figure 10 sensors-23-08380-f010:**
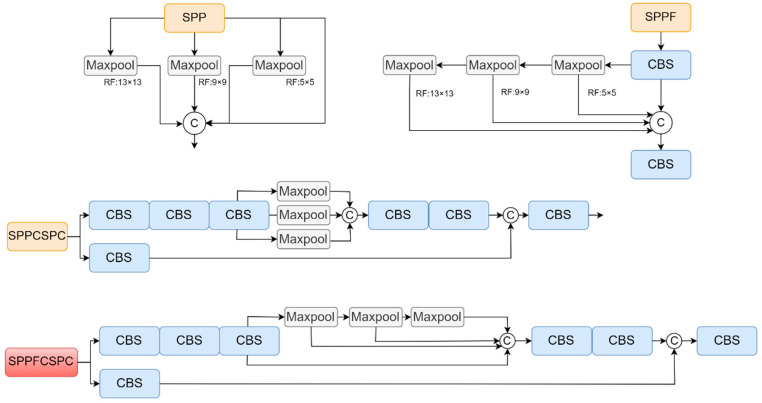
SPP, SPPF, SPPCSPC, and SPPFCSPC.

**Figure 11 sensors-23-08380-f011:**
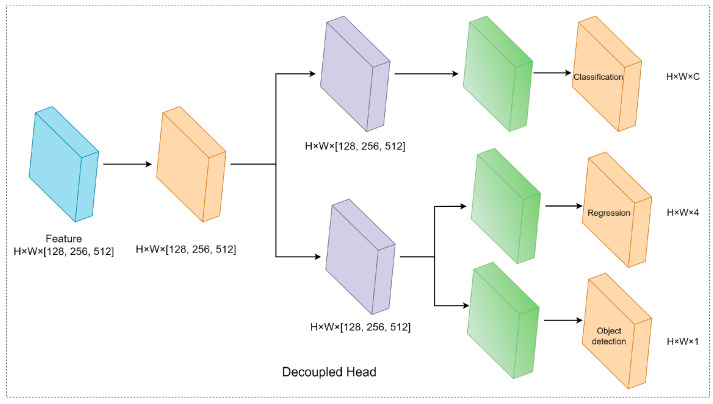
Decoupled head.

**Figure 12 sensors-23-08380-f012:**
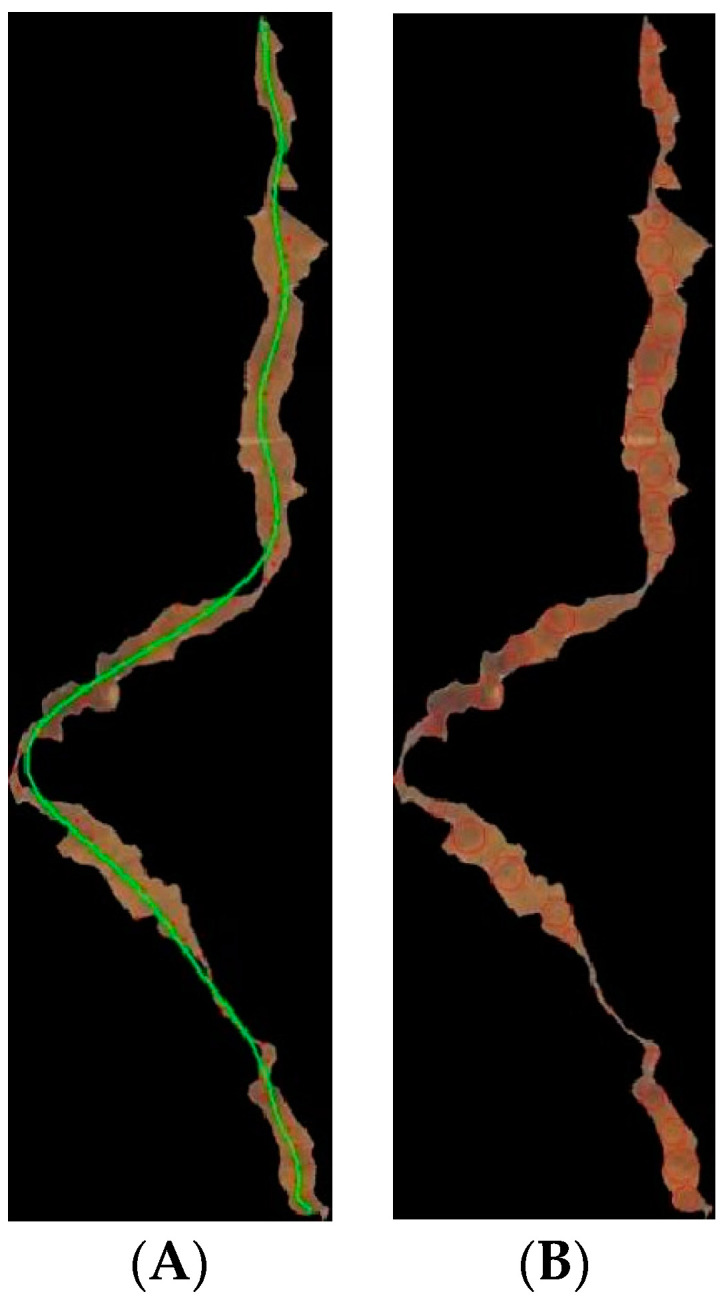
LWC algorithm schematic. (**A**) Tobacco shred length calculation. (**B**) Tobacco shred width calculation.

**Figure 13 sensors-23-08380-f013:**
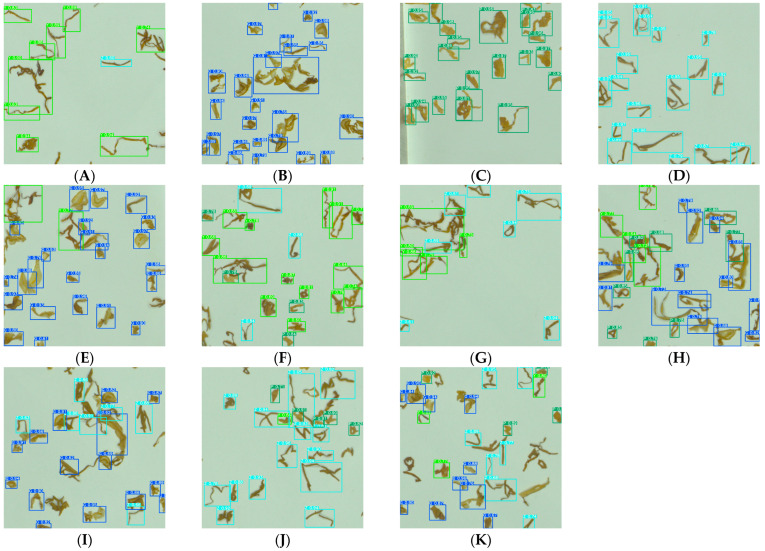
Eleven types of tobacco shred testing chart. (**A**) Pure tobacco silk. (**B**) Pure cut stem. (**C**) Pure expended tobacco shreds. (**D**) Pure reconstituted tobacco shred. (**E**) Tobacco silk-Cut stem. (**F**) Tobacco silk-Expended tobacco silk. (**G**) Tobacco silk-Reconstituted tobacco shred. (**H**) Cut stem-Expended tobacco silk. (**I**) Cut stem-Reconstituted tobacco shreds. (**J**) Expended tobacco silk-Reconstituted tobacco shred. (**K**) Tobacco silk-Cut stem-Expended tobacco silk-Reconstituted tobacco shred.

**Figure 14 sensors-23-08380-f014:**
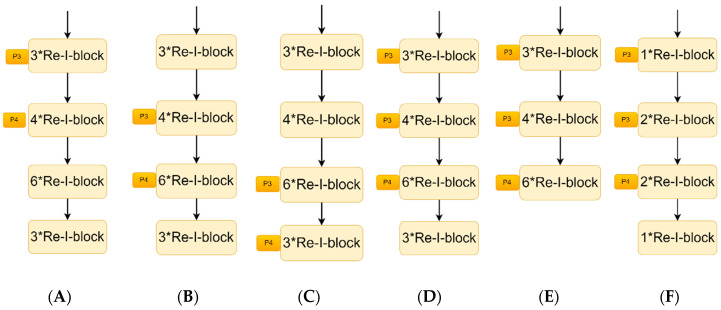
Different backbone networks. (**A**) B1. (**B**) B2. (**C**) B3. (**D**) B4. (**E**) B5. (**F**) B6.

**Figure 15 sensors-23-08380-f015:**
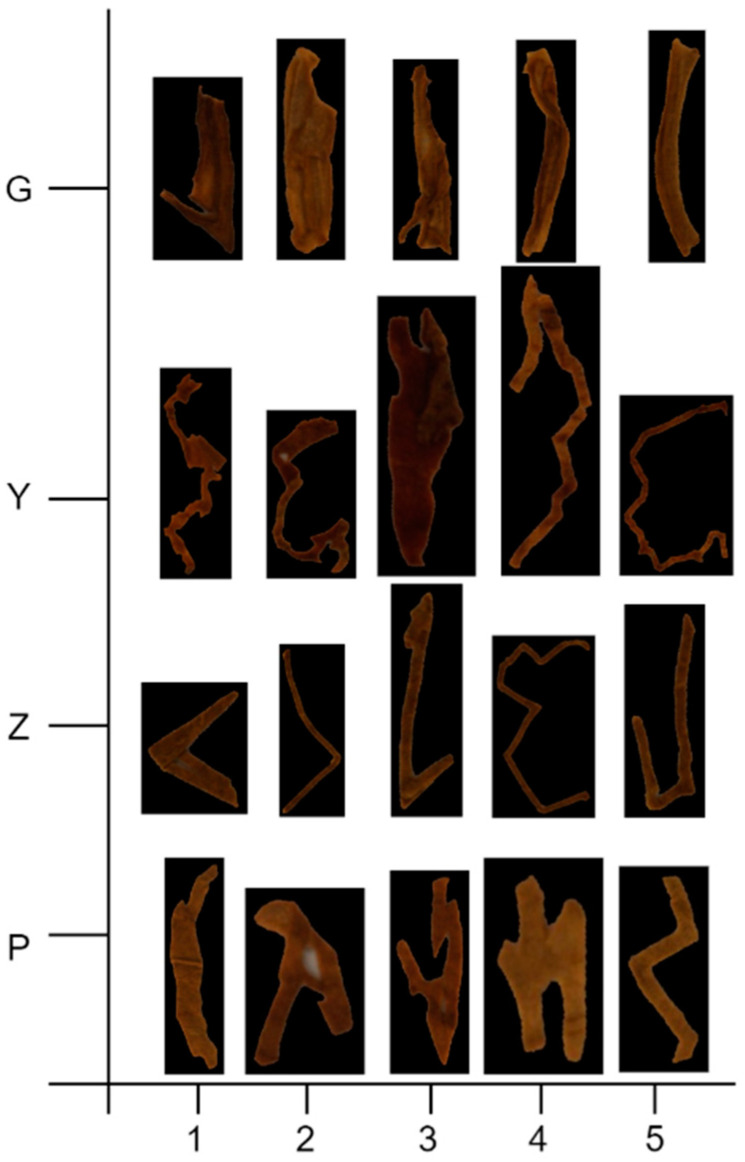
The original tobacco shred sample datasets. G stands for cut stem, Y for tobacco silk, Z for reconstituted tobacco shred and P for expended tobacco silk.

**Table 1 sensors-23-08380-t001:** Tobacco shred images shot.

No.	Corresponding to Figure 2	Construction Sources	Quantity
1	A	Pure tobacco silk	1000
2	B	Pure cut stem	1000
3	C	Pure expended tobacco silk	1000
4	D	Pure reconstituted tobacco shred	1000
5	E	Tobacco silk-Cut stem	200
6	F	Tobacco silk-Expended tobacco silk	200
7	G	Tobacco silk-Reconstituted tobacco shred	200
8	H	Cut stem-Expended tobacco silk	200
9	I	Cut stem-Reconstituted tobacco shred	200
10	J	Expended tobacco silk-Reconstituted tobacco shred	200
11	K	Tobacco silk-Cut stem-Expended tobacco silk-Reconstituted tobacco shred	100
Total			5300

**Table 2 sensors-23-08380-t002:** Tobacco shred dataset 1.

Dataset 1	Number of Training Sets	Number of Test Sets
Y	700	300
G	700	300
P	700	300
Z	700	300
Total	2800	1200

**Table 3 sensors-23-08380-t003:** Tobacco shred dataset 2.

Dataset 2	Number of Training Sets	Number of Test Sets
Blended Y	1190	510
Blended G	1190	510
Blended P	1190	510
Blended Z	1190	510
Total	4760	2040

**Table 4 sensors-23-08380-t004:** Parameters used in image enhancement.

Enhancement Method	Hyperparameter Values
Hsv (hsv_h, hsv_s, hsv_v)	0.015, 0.7, 0.4
Translate	0.2
Scale	0.5
Mosaic	1
Mixup	0.05
Paste_in	0.05

**Table 5 sensors-23-08380-t005:** Improved YOLOv7-tiny performance.

Model	Precision	Recall	mAP@.5	mAP@.5:.95	Parameters
YOLOv7-tiny-dataset 1	0.901	0.907	0.956	0.831	6,022,129
YOLOv7-tiny-dataset 2	0.834	0.802	0.889	0.736	6,022,129
Improved YOLOv7-tiny-dataset 2	0.883	0.853	0.932	0.795	20,572,002

**Table 6 sensors-23-08380-t006:** Performance comparison of different backbones.

Model	Precision	Recall	mAP@.5	mAP@.5:.95	Parameters
YOLOv7-tiny	0.834	0.802	0.889	0.736	6,022,129
YOLOv7-tiny-Regnet	0.849	0.792	0.886	0.724	7,416,313
YOLOv7-tiny-Efficient	0.867	0.797	0.898	0.738	7,934,413
YOLOv7-tiny-MobileNet	0.859	0.806	0.901	0.736	4,484,209
YOLOv7-tiny-Resnet50	0.864	0.825	0.913	0.766	28,056,977
YOLOv7-tiny-B1	0.863	0.832	0.911	0.763	27,365,009
YOLOv7-tiny-B2	0.871	0.845	0.920	0.774	27,446,929
YOLOv7-tiny-B3	0.868	0.835	0.916	0.77	27,610,769
YOLOv7-tiny-B4	0.879	0.847	0.926	0.781	27,467,537
YOLOv7-tiny-B5	0.859	0.827	0.908	0.768	12,398,353
YOLOv7-tiny-B6	0.876	0.845	0.924	0.777	13,259,537

**Table 7 sensors-23-08380-t007:** Performance comparison of different necks.

Model	Precision	Recall	mAP@.5	mAP@.5:.95	Parameters	Predicted Time (ms)
YOLOv7-tiny-B6	0.876	0.845	0.924	0.777	13,259,537	3.61
YOLOv7-tiny-B6-SPPFCSPC	0.877	0.846	0.923	0.776	13,259,537	2.55

**Table 8 sensors-23-08380-t008:** Performance comparison of different head.

Model	Precision	Recall	mAP@.5	mAP@.5:.95	Parameters	Predicted Time (ms)
YOLOv7-tiny-BS	0.877	0.842	0.923	0.774	13,259,537	2.55
YOLOv7-tiny-BS-decoupled-head	0.883	0.852	0.932	0.795	20,572,002	4.12

**Table 9 sensors-23-08380-t009:** Comparison of different object detection methods.

Model	Precision	Recall	mAP@.5	mAP@.5:.95	Parameters
Faster RCNN	0.753	0.781	0.834	0.693	41,808,406
RetinaNet	0.792	0.753	0.848	0.702	32,263,304
SSD	0.642	0.721	0.771	0.628	15,497,998
YOLOv7-tiny	0.834	0.802	0.889	0.736	6,022,129
Improved YOLOv7-tiny	0.883	0.853	0.932	0.795	20,572,002

**Table 10 sensors-23-08380-t010:** Comparison of different methods for tobacco shred detection.

Model	Detection of Tobacco Shred Type		Precision (%)	Parameters	Preprocess Time (s)	Start-Up Time (s)	Predict Time (s)	Total Time (s)
	Single	Overlapped						
Resnet50 (Zhong et al., 2021 [6])	√		94.32	25,636,712	×	2.60	33.16	35.76
Light-VGG (Niu et al., 2022 [7])	√		95.55	4,690,726	3.91	2.52	27.32	33.75
MS-Resnet (Niu et al., 2022 [2])	√		96.56	26,300,968	3.9	2.66	34.41	40.97
Ince-Resnet (Liu et al., 2022 [8])	√		97.23	34,353,336	23.53	2.66	40.98	67.17
Mask RCNN (Wang et al., 2023 [9])		√	89.1	27,763,373	×	5.99	69.05	75.04
Improved YOLOv7-tiny	√	√	93	20,572,002	1.32	3.32	0.17	3.49

√ indicates the ability to handle both single and overlapping tobacco.

**Table 11 sensors-23-08380-t011:** Size measurement results comparison.

Sample	Actual Length of Tobacco Shred (mm)	Actual Width of Tobacco Shred (mm)	Relative Error of Detected Length (%)	Relative Error of Detected Width (%)
Standard	3.000	9.000	−0.7	−0.8
Y1	6.36, 12.16, 10.16, 6.06	2.03, 2.30, 2.11, 2.03	−6.8	8.2
Y2	8.63, 10.24, 7.32, 5.32	2.13, 1.52, 1.63, 1.32	2.7	11.3
Y3	13.11, 3.34	2.91, 1.87	11.5	12.1
Y4	4.32, 15.32, 9.53, 7.21	0.87, 1.04, 1.32, 0.97	−1.3	10.9
Y5	13.07, 4.34, 4.58, 30.48	1.03, 0.87, 0.92, 1.04	5.9	−1.6
G1	14.78, 5.14	2.59, 0.85	−4.1	59.9
G2	18.22	3.9	1.9	−10.3
G3	21.28, 4.10	2.41, 1.55	5.5	41.9
G4	12.46, 7.49	2.56, 2.20	11.7	−2.1
G5	12.35, 10.94	2.3, 2.03	−0.3	3
P1	11.66, 13.21	2.66, 2.23	0.5	4.7
P2	6.57, 11.08	1.91, 1.67	−5.3	32.4
P3	12.38, 15.57	1.74, 1.76	−3.4	−3.4
P4	7.95, 8.51	1.32, 1.11	−8.3	47.3
P5	7.75, 4.79, 4.09	1.22, 1.03, 1.20	−14.6	8.7
Z1	10.16, 7.4	1.29, 2.4	−13.7	34.4
Z2	13.21, 20.12	1.05, 1.02	−1.9	2.4
Z3	15.69, 4.04	1.12, 1.06	−1.4	4.6
Z4	12.33, 12.26, 8.79, 16.89	0.86, 0.94, 0.91, 0.88	−4.4	−0.8
Z5	14.46, 3.99, 6.56	1.11, 1.07, 0.83	−8.8	−0.2

**Table 12 sensors-23-08380-t012:** Average measurement results of 20 tobacco shred samples.

Sample	Relative Error of Detected Length (%)	Relative Error of Detected Width (%)
Total-Y	2.4	8.18
Total-G	2.94	18.48
Total-P	−6.22	17.94
Total-Z	−6.04	8.08
Total	−1.7	13.2

## Data Availability

The data that support the findings of this study are available from the corresponding author upon reasonable request.

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
