# Peer review of "A New Efficient Multi-Object Detection and Size Calculation for Blended Tobacco Shreds Using an Improved YOLOv7 Network and LWC Algorithm"

_sensors, 2023, doi:10.3390/s23208380_

Round 1

Reviewer 1 Report

1.     The abstract and conclusion need to be improved. The abstract must be a concise yet comprehensive reflection of what is in your paper. Please modify the abstract according to “motivation, description, results and conclusion” parts. I suggest extending the conclusions section to focus on the results you get, the method you propose, and their significance.

2.     What is the motivation of the proposed method? The details of motivation and innovations are important for potential readers and journals. Please add this detailed description in the last paragraph in section I. Please modify the paragraph according to "For this paper, the main contributions are as follows: (1) ......" to Section I. Please give the details of motivations. In Section 1, I suggest the authors can amend your contributions of manuscript in the last of Section 1.

3.     The description of manuscript is very important for potential reader and other researchers. I encourage the authors to have their manuscript proof-edited by a native English speaker to enhance the level of paper presentation. There are some occasional grammatical problems within the text. It may need the attention of someone fluent in English language to enhance the readability.

1.     Please give the details of proposed method for proposed model. I suggest the authors amend the calculation of your size of proposed method and the details is important for proposed method.

2.     The content of experiments needs to amend related experiments to compare related SOTA in recent three years. I recommend the authors amend related experimental results of proposed method of SOTA according to the published paper in IEEE, Springer and Elsevier.

3.     However, the manuscript, in its present form, contains several weaknesses. Adequate revisions to the following points should be undertaken in order to justify recommendation for publication.

4.     In the conclusion section, the limitations of this study and suggested improvements of this work should be highlighted.

5.     Provide a critical review of the previous "journal" (not conference) papers in the area and explain the inadequacies of previous approaches.

6.     I suggest the authors revise Section 1 and Section 2. Please revise the content according to the development of timeline.

None

Author Response

Comments 1: The abstract and conclusion need to be improved. The abstract must be a concise yet comprehensive reflection of what is in your paper. Please modify the abstract according to “motivation, description, results and conclusion” parts. I suggest extending the conclusions section to focus on the results you get, the method you propose, and their significance.

Reply 1: Thank you for suggestion. We have revised the abstract and conclusion as requested in the revised manuscript.

Comments 2: What is the motivation of the proposed method? The details of motivation and innovations are important for potential readers and journals. Please add this detailed description in the last paragraph in section I. Please modify the paragraph according to "For this paper, the main contributions are as follows: (1) ......" to Section I. Please give the details of motivations. In Section 1, I suggest the authors can amend your contributions of manuscript in the last of Section 1.

Reply 2: We have added detailed description in the last paragraph in section I and revised the section I and contribution as requested in the revised manuscript.

Comments 3: The description of manuscript is very important for potential reader and other researchers. I encourage the authors to have their manuscript proof-edited by a native English speaker to enhance the level of paper presentation. There are some occasional grammatical problems within the text. It may need the attention of someone fluent in English language to enhance the readability.

Reply 3: Thank you for suggestion. We have manuscript proof-edited as requested.

Comments 4: Please give the details of proposed method for proposed model. I suggest the authors amend the calculation of your size of proposed method and the details is important for proposed method.

Reply 4: We have revised the description of size calculation of proposed method as requested.

Comments 5: The content of experiments needs to amend related experiments to compare related SOTA in recent three years. I recommend the authors amend related experimental results of proposed method of SOTA according to the published paper in IEEE, Springer and Elsevier.

Reply 5: Thank you for pointing this out. We consulted articles on object detection in these journals of IEEE, Springer and Elsevier in the past three years. Among similar target detection needs, YOLO is the most widely used network Model, including Faster RCNN, RetinaNet, SSD, Global Local detection Model, and Cascade R-CNN. The target detection network is divided into single-stage detection and two-stage detection. In this paper, a single level target detection network is selected considering timeliness. Currently, single-phase target detection networks include YOLO, SSD and Retinanet. Two-stage target detection networks include :RCNN, Fast RCNN, Faster RCNN and Cascade RCNN. The comparison networks used in this paper include :Faster RCNN, RetinaNet and SSD, which involve single-level target detection network and two-level target detection network. Global local detection models are difficult to replicate and compare in self-built networks. Both Cascade R-CNN and Faster RCNN are two-stage networks, but Faster RCNN has better timeliness, so Faster RCNN is chosen.

Comments 6: In the conclusion section, the limitations of this study and suggested improvements of this work should be highlighted.

Reply 6: Thank you for suggestion. In the conclusion section, we have revised the limitations of this study and the improvements of this work.

Comments 7: Provide a critical review of the previous "journal" (not conference) papers in the area and explain the inadequacies of previous approaches.

Reply 7: Thank you for suggestion. Previous "journal" (not conference) papers in the area specifically include:

(1)Zhong, Y., Zhou, M., Xu, Y., Liu D., Wang, H., Dong, H., ... & Xing, J. (2021). A method for identifying types of tobacco strands based on residual neural network. Tobacco Science & Technology(05), 82-89. doi:10.16135/j.issn1002-0861.2020.0602. (journal) The network model used in this paper is Resnet50.

(2) Niu, Q., Yuan, Q., Jin, Y., Wang, L., & Liu, J. (2022). ldentification of tobacco strands types based on improved VGG16 convolutional neural network. Foreign Electronic Measurement Technology(09), 149-154. doi:10.19652/j.cnki.femt.2203982. (journal) The network model used in this paper is Light-vgg.

(3) Niu, Q., Liu, J., Jin, Y., et al., 2022. Tobacco shred varieties classification using Multi-Scale-X-ResNet network and machine vision. Frontiers in plant science, 13. (journal) The network model used in this paper is MS-Resnet.

(4) Liu, J., Niu, Q., Jin, Y., Chen, X., Wang, L., & Yuan, Q. (2022). Research on Tobacco Shred lmage Recognition Method Based on Efficient Channel Attention Mechanism and Multi-Scale Feature Fusior. Journal of Henan Agricultural Sciences(11), 145-154. doi:10.15933/j.cnki.1004-3268.2022.11.017. (journal) The network model used in this paper is Ince-Resnet.

(5) Wang, L., Jia, K., Fu, Y., Xu, X., Fan, L., Wang, Q., ... & Niu, Q. (2023). Overlapped tobacco shred image segmentation and area computation using an improved Mask RCNN network and COT algorithm. Frontiers in Plant Science, 14. (journal) The network model used in this paper is Mask RCNN.

In the above research process, the main ideas of tobacco shred recognition are divided into: 1) Firstly, a single tobacco shred image containing only multiple single non-overlapped cases is processed by segmentation algorithm to establish a single tobacco shred object image. Secondly, the image classification methods are performed on multiple single tobacco shred images using image classification models such as Resnet50, Light-vgg, MS-Resnet and Inception-Resnet.

2) Firstly, the single tobacco shred image in the case of containing multiple overlapped tobacco shreds is processed by a segmentation algorithm to identify overlapped tobacco shred. Secondly, a segmentation model is used for a single overlapped tobacco shred object to perform image segmentation processing, such as Improved Mask RCNN.

3)  In this paper, the idea is to create blended tobacco shred image that reflects the real field situation as a sample data set, and one tobacco shred image contains both single and overlapped tobacco shreds, which is directly processed by image tagging, without need to create multiple single tobacco shred images in advance through segmentation algorithms, and directly output the corresponding blended tobacco shred detection results with multi-object detection model.

Comments 8: I suggest the authors revise Section 1 and Section 2. Please revise the content according to the development of timeline.

Reply 8: Thank you for suggestion. I have sorted the references according to the timeline, with specific changes in lines 52 and 88 through 119.

Reviewer 2 Report

Dear Authors,

Below are detailed comments on the manuscript:

1) In "Abstract" specify precisely the purpose of the work

2) Figure 1 Tobacco shred image acquisition device - the description of the stand's elements is not legible,

3) Figure 6, 7, 11 - note as above,

4) Add more information about the loss function; in non-linear regression and algorithms this is the basis.

5) Was there a need to scale the model? If not, include it in your summary.

6) 5 Conclusion - the chapter is a summary, I suggest changing it to "Summary",

7) 5 Conclusion - font

Author Response

Comments 1: In "Abstract" specify precisely the purpose of the work.

Reply 1:  We have revised the abstract as requested in the revised manuscript.

Comments 2: Figure 1 Tobacco shred image acquisition device - the description of the stand's elements is not legible.

Reply 2: Thank you for your suggestion. We have revised Figure 1 as requested.

Comments 3: Figure 6, 7, 11 - note as above.

Reply 3: Thank you for your suggestion. We have revised Figure 6, 7, 11 as requested.

Comments 4: Add more information about the loss function; in non-linear regression and algorithms this is the basis.

Reply 4: Thank you for pointing this out. I have, accordingly, revised this on line 479.

Comments 5: Was there a need to scale the model? If not, include it in your summary.

Reply 5:  In the actual application process, the convenient deployment of the model is very critical. Models with too many parameters are difficult to deploy and costly. From this consideration, we scaled the model.

Comments 6: 5 Conclusion - the chapter is a summary, I suggest changing it to "Summary". Reply 6: Thank you for pointing this out. I have, accordingly, revised this on line 675.

Comments 7: 5 Conclusion – font.

Reply 7: I have, accordingly, revised it in the revised manuscript.

Reviewer 3 Report

A new efficient multi-object detection and size calculation for 2 blended tobacco shred using an improved YOLOv7 network 3 and LWC algorithm.

This study focuses on the two challenges of identifying 19 blended tobacco shred with single tobacco shreds and overlapped tobacco simultaneously in the 20 field application and calculating the unbroken tobacco shred rate. In this paper, a new multi-object 21 detection model is developed for blended tobacco shred images based on an improved YOLOv7- 22 tiny. This research is good, but I still have some comments for improvement.

1. Pleased revised all figure with a good resolution.

2. It is suggest to add literature review about YoloV8. Why the author chose Yolov7?

3. Explain all formula in this manuscript.

4. Explain more about dataset that used in the experiment.

5. It is suggest to test the proposed method to the other dataset.

6. All class is too similar Table 1 and Figure 2. How the author solved this problem? Explain in detail so the result will be convincing.

7. Add related reference:

DOI: 10.3390/app122211318

DOI: 10.3390/bdcc7010053

DOI: 10.3390/agriculture12101659

DOI: 10.3390/bdcc7020094

DOI: 10.3390/drones7050304

DOI: 10.3390/electronics12102323

DOI: 10.3390/s23031347

DOI: 10.3390/agronomy13040999

Author Response

Comments 1: Pleased revised all figure with a good resolution.

Reply 1: Thank you for suggestion. All images have been proved to a higher  resolution in the revised manuscript.

Comments 2: It is suggested to add literature review about YoloV8. Why the author chose Yolov7?

Reply 2: In 2022 , we implemented the detection model and algorithm for tobacco shred components detection and the manuscript has finished. At that time, the latest object detection algorithm in the YOLO family was YOLOv7. We will carry out the research of YOLOv8 in the future.

Comments 3: Explain all formula in this manuscript.

Reply 3: Thank you for suggestion. You can check on lines 415, 465, 467, 469, 470, 480, 484, 486 and 487 in the revised manuscript.

Comments 4: Explain more about dataset that used in the experiment.

Reply 4: We shot and established two types of original blended tobacco shred image datasets, 4,000 non-overlapped tobacco shreds consisting of images captured from four tobacco shred varieties and 5,300 blended tobacco shreds consisting of images captured from both four non-overlapped tobacco shred varieties and seven types of overlapped tobacco shred. Dataset1 is established for training base model and dataset2 is to mimic the actual field and increase the robustness of the model.

Comments 5: It is suggested to test the proposed method to the other dataset.

Reply 5: Thank you for suggestion. The datasets based tobacco shred images shot by our team have been applied in the target detection for a certain cigarette brand. Next, we will build more tobacco shred datasets from different cigarette brands to test the proposed method adaptability.

Comments 6: All class is too similar Table 1 and Figure 2. How the author solved this problem? Explain in detail so the result will be convincing.

Reply 6: Thank you very much for your suggestions. The tobacco shred in cigarettes consists of four varieties of tobacco silk, cut stem, expended tobacco silk and reconstituted tobacco shred, as shown in Figure 1. A, B, C and D respectively. The four types of tobacco shred are very similar, but there are still some tiny differences.

Figure 1 | Four types of tobacco shred. (A) Tobacco silk. (B) Cut stem. (C) Expended tobacco silk. (D) Reconstituted tobacco shred.

  1. Tobacco silk: dark yellow in color, mostly slender filaments, with some veins visible in the texture.
  2. Cut stem: the color is bright yellow, and the stalk structure is clearly visible in the texture.
  3. Expended tobacco silk: the color is bright yellow, the edge is smooth, because the veins are not visible in the texture after the expansion treatment, the surface is relatively flat.
  4. Reconstituted tobacco shred: the color is dark yellow, mostly slender filaments, the edge is not smooth with burrs, and filaments are obviously visible in the texture.

Through the above description, the four types of tobacco shred are tiny different in the color and texture. It is indeed difficult to classify tiny single tobacco shreds with complex morphological characteristics and further classifying tobacco shreds with 24 overlapped types. Especially, tobacco silk are difficult to distinguish from expanded tobacco shred because the expanded tobacco shred is obtained by the expansion treatment of the tobacco silk. It is a challenge task to detect blended tobacco shreds with complex tobacco shred forms containing both non-overlapped and overlapped tobacco shreds under a single picture efficiently and accurately. So, we researched many models and did many improvement work according to extract tiny different features among them to obtain a better detection effect.

Comments 7: Add related reference:

DOI: 10.3390/app122211318

DOI: 10.3390/bdcc7010053

DOI: 10.3390/agriculture12101659

DOI: 10.3390/bdcc7020094

DOI: 10.3390/drones7050304

DOI: 10.3390/electronics12102323

DOI: 10.3390/s23031347

DOI: 10.3390/agronomy13040999

Reply 7: The above related references have been added in lines 89, 89, 89, 89, 112, 116, 89, and 107 in the revised manuscript.

Round 2

Reviewer 1 Report

None

none

Author Response

Dear editor:

Thank you very much for taking the time to review this manuscript. 

Sincerely yours,

Kunming Jia

Reviewer 3 Report

All class is too similar Table 1 and Figure 2. How the author solved this problem? Explain in detail so the result will be convincing. 

The author need to add more discussion about this.

Author Response

Comments 1: All class is too similar Table 1 and Figure 2. How the author solved this problem? Explain in detail so the result will be convincing.

Reply 1: The target detection model is divided into one-stage detection model and two-stage detection model. In order to meet the actual field application, the object detection model in this paper is very strict on real-time requirements, so the one-stage detection model with very fast detection speed is chosen. One-stage detection models include :YOLO, SSD and RetinaNet. Among them, YOLOv7 in YOLO series has better detection accuracy and timeliness. YOLOv7 performs rapid target detection on blended tobacco shred, mainly including the following processes:

1) Firstly, input the blended tobacco shred images to the backbone of the YOLOv7 network. Backbone consists of multiple CBS modules. The CBS consists of a convolutional layer, a BN layer and a Silu layer. Backbone extracts the features of blended tobacco shred images and extracts the unique features (color and shape) of each of the four types of tobacco shred.

2) Secondly, Neck structure is similar to Panet. The shallow and deep features of blended tobacco shreds were fused to form a feature map with multiple information, which enhanced the characterization ability of the model.

3) Finally, the head structure is used to objectness, class and bbox of the generated feature map of multiple information.

Considering the complex and similar characteristics of the four tobacco shreds and the small target of the tobacco shred fragments, the basic YOLOv7 is not suitable for the rapid and accurate detection of blended tobacco shred. An improved YOLOv7 model is proposed, which improves multi-scale convolution, feature reuse, different PANet join and compression models. Resnet19 was constructed as the new backbone of YOLOv7-tiny to enhance the depth and width of the model feature map without significantly increasing the network parameters, and enhance the unique feature extraction capability of the model for the four types of tobacco shred. Secondly, SPPCSPC in Neck structure is changed to SPPFCSPC to improve the inference speed of model without increasing model parameters. Finally, the Head structure is optimized as a decoupled Head, and the convolution of detection tasks is decoupled, which improves the feature graph utilization ability of the model and the multi-target detection performance of the model. Finally, by comparing different target detection models and different mixed tobacco detection algorithms, the improved YOLOv7 has the best performance and can quickly and accurately complete the multi-target detection of mixed tobacco in the actual field. The experimental results show that the final detection accuracy is mAP@.5, mAP@.5:. The test time was 0.883, 0.932, 0.795 and 4.12ms, respectively.